# Life Cycle Assessment Project for the Brazilian Egg Industry

**DOI:** 10.3390/ani13091479

**Published:** 2023-04-27

**Authors:** Fabiane de Fátima Maciel, Richard Stephen Gates, Ilda de Fátima Ferreira Tinôco, Fernanda Campos de Sousa, Nathan Pelletier, Maro A. Ibarburu-Blanc, Carlos Eduardo Alves Oliveira

**Affiliations:** 1Department of Agricultural Engineering, Universidade Federal de Viçosa, Viçosa 36570-900, MG, Brazil; iftinoco@ufv.br (I.d.F.F.T.); carloseoliveira@ufv.br (C.E.A.O.); 2Departments of Agricultural and Biological Engineering, and Animal Science, Iowa State University, Ames, IA 50010, USA; rsgates@iastate.edu (R.S.G.); maro@iastate.edu (M.A.I.-B.); 3Faculty of Management, University of Britishi Columbia, Okanagan Campus, Kelowna, BC V0A-V9Z, Canada; nathan.pelletier@ubc.ca

**Keywords:** LCA, LCI, LCIA, agricultural sector, egg production, intensive system, sustainability

## Abstract

**Featured Application:**

**The topic addressed is highly relevant to the Brazilian Egg Production Industry.**

**Simple Summary:**

Promoting sustainability in food production has become fundamental to meeting the demands of the market, mainly because it presents a trend of expansion around the world. Life cycle assessment is a recognized methodology for providing quantitative information on environmental impacts caused throughout the production cycle in different categories. With the aim of providing transparent information to Brazilian producers and consumers about the impacts generated in egg production, this review presents the LCA methodology addressed in international studies based on the ISO 14040 and 14044:2006 standards. The results generally point to feed production and supply as the main point of impact on production, covering all impact categories, including Acidification, Eutrophication, Fossil Fuel Depletion, Global Warming Potential, Ozone Layer Depletion and Ecotoxicity. After quantifying emissions, it becomes possible to estimate the value of emissions per unit of eggs produced. The results obtained during the analysis will be able to promote good environmental practices and new ecological strategies, including animal welfare, food safety, the rational use of natural resources, the reduction in gas emissions and the generation of data on future scenarios.

**Abstract:**

Brazil is among the ten largest egg producers in the world. The domestic consumption of Brazilian eggs is 99.6%, the rest being exported to more than 82 countries, with an expectation of growth in the foreign market. The Brazilian egg industry has evolved considerably in recent decades, incorporating new technologies and smart practices. However, there is no assessment of how production could become more sustainable over the years. The LCA (Life Cycle Assessment) approach aims to recognize the polluting potential, identify the environmental impacts generated and reduce these impacts throughout production. On a global scale, researchers approach LCA as a constructive and quantitative technique, and there is great interest in implementing an LCA for the Brazilian egg production sector. With the aim of introducing the LCA methodology to the Brazilian egg industry, this review presents the concept and accounting structure of LCA through LCI (Life Cycle Inventory) and LCIA (Life Cycle Impact Assessment), based on the ISO 14040 and 14044:2006 standards, to quantify the environmental performance of production and identify areas for future improvement, thus promoting the environmental footprint of the Brazilian egg industry.

## 1. Introduction

According to data from the FAO (Food and Agriculture Organization) [1], the world population is expected to grow to 9.7 billion by 2050. The challenges are not limited to providing more food for a growing population but also include social, economic and environmental challenges [2]. In 2021, world egg production exceeded 86.4 million tons, compared to 87.1 million tons produced in 2020. This number represents the resumption of economic growth caused by the second year of the Covid-19 pandemic, a constant observed in Brazil and in the world [3,4]. However, the historical series of world egg production has increased by more than 100% from 1990 to the present day [3].

Brazil is among the ten largest egg producers in the world, occupying the fifth position in the world ranking, in the year 2022 [5]. According to the ABPA (Brazilian Animal Protein Association) [6], 99.6% of Brazilian egg production is destined for the national market. Only 0.4% is destined for the international market (with growth expected for the coming years), exported to more than eighty countries distributed in the regions of Africa, America, Asia, Europe, Oceania, the Middle East and the European Union. They are in the ranking of the five largest importers of Brazilian eggs in the year 2022: United Arab Emirates (6916 tons), Japan (1171 tons), Qatar (486 tons), United States of America (472 tons) and Oman (408 tons).

According to IBGE (Livestock Production Statistics Report) [7], the production of chicken eggs reached 1.03 billion dozens in the fourth quarter of 2022. The result represented an increase of 3.1% compared to the same period of the previous year and an increase of 1.3% compared to the third quarter of 2022 [7]. This number corresponds to the increase in consumption in the domestic market and the continuous growth of the production of Brazilian eggs [6], which have remained on the market because they are the lowest-priced source of animal protein and, therefore, more accessible to all social classes [4]. Some of the main challenges of modern production include promoting sustainability in the production process, contributing to the reduction in greenhouse gas emissions, the loss of biodiversity and the depletion of finite natural resources. For future sustainable poultry production, it is essential that these challenges are overcome [8].

The impact of different egg production systems is still considered a significant issue, including housing types, health, welfare and animal behavior. The intensive cage-based production system has become a subject of debate among advocates of animal welfare and protection, as well as among researchers and producers. However, in addition to animal welfare, there are many other aspects involving sustainability, including economics, environmental factors, human health, food safety and social values [9].

The predominant egg production system in Brazil is intensive, with conventional cages in sheds. It is estimated that the cage system corresponds to 95% of the total egg production. In this production system, two types of installation predominate: pyramidal, corresponding to 64%, also known as the Californian model; and vertical, corresponding to 36%, both systems differing only in the arrangement of the cages [10].

Brazilian commercial egg producers comply with regulations that address different specifications and information related to production, complementing the safety and compliance aspects of eggs distributed in the market [8]. The main regulatory bodies are the MAPA (Ministry of Agriculture and Livestock) [11], ANVISA (National Health Surveillance Agency) [12] and INMETRO (National Institute of Metrology, Quality and Technology) [13,14]. In compliance with the standards, guiding documents obtained by EMBRAPA (Brazilian Agricultural Research Corporation) [11] and ABPA (Brazilian Association of Animal Protein) [12] address the stages of production, from the origin of inputs to the final product for the consumer. In 2009, ABNT (Brazilian Association of Technical Standards) published the following NBR (Brazilian Technical Standard) as Portuguese versions: NBR ISO 14040:2009—Life Cycle Assessment: Principles and Structure [15]; and NBR ISO 14044:2009—Life Cycle Assessment: Requirements and Guidelines [16].

The agricultural sector undergoes constant changes related to economics, technology and social, environmental and marketing aspects, which occur simultaneously under different conditions and at high speeds [17]. Thus, solutions are needed for some important issues, such as the availability of natural resources, the control of the generation of pollutants and the extraction of sustainable raw materials. The main strategies used by producers are the intensification of the use of available resources and investing in management training, considering that the success in the productive activity is a function of the producer’s capacity to adhere to the technological tools available [18].

Because it is an intensive production system with growing demand, it is important to carry out studies that indicate clearly and precisely how much production impacts the environment. Every production process generates the depletion of natural resources, material flows, energy expenditure and increased emissions along the production chain. Another important aspect is the specific assessment of the impacts of this chain. From a broader perspective, all inputs should be analyzed with the objective of predicting mitigation practices and proposed solutions. However, there is a lack of studies on production scenarios and coefficients that integrate the environmental impacts resulting from the inefficiency in the use of inputs in egg production [19].

Initially, the LCA (Life Cycle Assessment) was carried out as an environmental tool to be applied in industry in 1960 in Europe [19]. In recent years, LCA has been used in the agricultural sector in response to the growing demand for information about production and its production chain [20]. However, there are still challenges, such as encouraging producers to use LCA as a sustainable production tool as a way to guarantee the reduction in impacts and the generation of future profits. The LCA approach facilitates the identification of opportunities for improvement and resource efficiency, together with the purpose of reducing the emissions, acknowledging of the potential load transfer in different types of impacts and/or different stages of the supply chain and proposing results for decision makers. The LCA tool provides a basis for sustainable interventions, analyzing the main variables of the supply chain [20].

In view of the above, the objective of this review is to present, in general, the environmental LCA approach of an egg farm, from the cradle to the gate, according to ISO 14040 and 14044:2006 [21,22], in order to identify which tools are available so that this assessment can promote the environmental footprint of the egg industry in Brazil.

## 2. Life Cycle Assessment (LCA)

### 2.1. Conceptualization

With the growing awareness of the importance of environmental protection and the possible actions associated with products, both manufactured and consumed, it has become necessary to develop methods for better understanding and addressing such impacts. One of the techniques under development for this purpose is the LCA, according to ISO 14040 and 14044:2006 [21,22].

These standards [21,22] present LCA as a technique for assessing the potential environmental impacts associated with a product. This technique can also be considered a valuable tool for dealing with information about real impacts throughout the life cycle of products, from the acquisition of raw materials through production, classification, use, post-use treatment, recycling and final disposal. The product life cycles involves material, energy and economic flows. These, in turn, involve local impacts, consumers and all actors in the supply chain [23].

The LCA employs value judgments consistently and transparently and, in some cases, allows practitioners to make modeling choices based on their own values. As an example, one can cite the number of years into the future that environmental impacts should be considered in the assessment [24].

The LCA results of a product or process promote opportunities for improvement and environmental performance at various points in its life cycle. In this way, it is possible to assist decision makers in strategic planning and in the management of relevant environmental performance indicators [25,26]. It is also possible to contribute to marketing, through the implementation of an ecological labeling system, such as, for example, disclosing the environmental footprint of the product or process as a communication strategy [27]. However, the conclusion of an LCA allows companies to know how to quantify the sustainability of a product or process and which environmental aspects can be improved in order to allow for the reduction in potential environmental impacts [28].

LCA can be considered one of several environmental management techniques such as Environmental Risk Assessment, Environmental Performance Assessment, Environmental Audit and Environmental Impact Assessment, among others. However, the LCA does not address the economic or social aspects of a product [22]. According to ISO 14040 and 14044:2006 [21,22], LCA consists of four phases, starting with the definition of the objective and scope, moving on to an inventory analysis phase and impact assessment study and, finally, ending with the phase of interpretation of data, as illustrated in Figure 1.

The scope of an LCA depends directly on the intended object or use of the study. The depth and breadth of an LCA can vary, depending on the purpose of the study. An LCI (Life Cycle Inventory) analysis phase is the verification of the data from an inventory against the input/output of a system. This phase involves the collection of base data to achieve the objectives of the study in question. The LCIA (Life Cycle Impact Assessment) phase is the third phase of the LCA. The purpose of the LCIA is to provide additional information to assist in evaluating the results of a product system’s LCI category to better understand its environmental significance. Lifecycle interpretation is the final phase of the LCA procedure, in which the results of an LCI and/or an LCIA are summarized and discussed as a basis for carrying out recommendations and decision making in accordance with the objective definition and scope [22,25].

### 2.2. Accounting Structure

#### 2.2.1. Life Cycle Inventory (LCI)

As described in the ISO 14040 and 14044:2006 [21,22], an LCA is “a compilation and assessment of the environmental inputs, possibilities and impacts of a system product throughout its life cycle”. The LCI is considered a crucial second phase of LCA, as it deals with the quantification and accumulation of input data and the processes of a system. Thus, the LCI method chosen must comprise the calculation technique, its relative advantages and its limitations for the intended purpose [26].

By quantifying requirements such as energy and raw material consumption, atmospheric emissions, water consumption and solid waste generation, among other information, LCI directly interferes in the LCA of a product, process or activity [27]. The EPA (Environmental Protection Agency) documents (1993 and 1995) [27] define the four steps of a life cycle inventory: process flow diagram, data collection plan, data collection and outcome evaluation.

According to Suh and Huppes [28], there are six methods of compilation of the LCI (Life Cycle Inventory), namely: Process Flow Diagram, Product System Matrix Expression, LCI based on Input/Output, Layered Hybrid Analysis, Hybrid Analysis based on Input/Output and Integrated Hybrid Analysis. These authors concluded that, for LCA studies, an input/output LCI database is more available and developed in regionalized cases, linked (connected) to a local system.

To Islam et al. [26], the LCI has evolved significantly, becoming a more robust tool for sustainable practices. Different LCI methods imply distinct levels of complexity and data requirements. As there are many LCA software available on the market, the scientific validation of LCI methodologies is possible. The authors concluded that, in a faster ecological manufacturing decision, the LCI Input/Output is adequate; however, if some data related to the process are available in the Hybrid Input/Output database, these provide a better result.

Guinée et al. [29] considered the ISO 14040 and 14044:2006 [21,22] a biophysical accounting framework used to catalog the input materials of energy and natural resources that will provide emissions associated with each stage of the life cycle of a product. LCI describes, in terms of its quantitative contributions, a specific set of environmental impact categories.

The LCA database most used today in scientific studies is Ecoinvent [30], with about 4500 users in more than 40 countries, containing international lifecycle inventory data on energy supply, resource extraction, material origination, chemical products, metals, agriculture, waste management services and transportation services. Each dataset is provided as a unit process and aggregate system process. In addition, reports are published with information on modeling procedures and assumptions. The latest version is Ecoinvent v.3.7.1, with databases specifically adapted to OpenLCA [31].

Updating the Brazilian database was made possible by the ICVAgroBR project, coordinated by EMBRAPA Environmental [32] and funded by the SRI (Sustainable Recycling Industries) program of the Swiss government’s Secretariat for Economic Affairs. A total of 632 new datasets were integrated into the new version of Ecoinvent, including life cycle inventories of some of the main Brazilian agricultural products, contributing to the increase in their occurrence in the international market, which is increasingly demanding in terms of environmental aspects [33].

The IBICT (Brazilian Institute of Information in Science and Technology) [34], in partnership with the EMBRAPA Environment [32], promotes the structuring of the National Bank of Life Cycle Inventories of Brazilian Products [35]. According to Rodrigues [36], this database should reach 300 available inventories, mostly products from the agricultural chain. This author also claims that the inventory is produced from its initial phase; it becomes a slower and more costly process. If the inventory is available in a database, the authors will be able to carry out the analysis and generate studies without the need for complete data surveys. During the structuring of the agreement that culminated with the availability of Brazilian data in the Ecoinvent database [37], EMBRAPA [33] formalized the donation of data to the SICV Brazil (National Bank of Life Cycle Inventories) [38], managed by the IBICT [39]. In addition to international recognition, an update of the data will contribute to the practice of increasing access to national data among Brazilian professionals and researchers [33].

The inventory process can be considered a complete and more complex survey, which can generate environmental declarations of the product, which is another way of demonstrating its environmental performance [36]. When documenting the lifecycle inventory results, it is important to describe the entire methodology covered and define the applicable systems and thresholds that were adjusted and any assumptions made in carrying out the inventory analysis. The result of the inventory analysis is a list containing the amount of pollutants released into the environment and the amount of energy and materials consumed in the production process [27].

#### 2.2.2. Life Cycle Impact Assessment (LCIA)

The LCIA (Life Cycle Impact Assessment) phase is considered an assessment of the potential impacts related to human health and the environment identified during the LCI, contemplating the third phase of an LCA. The LCIA aims to provide an aggregation of inventory data using additional information, such as (internationally accepted) performance levels, to understand/translate the magnitude and importance of the results for impact assessment [23]. Life Cycle Data Interpretation is seen as a systematic technique for identifying, quantifying, verifying and evaluating information based on all previous results, such as those from LCI and LCIA [27]. ISO 14040 and 14044:2006 [21,22] defined the following two lifecycle interpretation objectives: (1) analyze the results, arrive at the consequences, explain the limitations, provide recommendations based on the process of the previous LCA phases and, finally, report the results of the life cycle interpretation transparently; and (2) provide a readily understandable, complete and consistent presentation of the LCA results, consistent with the purpose and scope of the study.

For an LCIA, several impact categories are selected according to the objective and scope defined in the study. According to Mendes et al. [40], traditional impact categories are defined by resource depletion, land use, climate change, stratospheric ozone depletion, human toxicity, aquatic ecotoxicity, terrestrial ecotoxicity, the formation of photo-oxidants, acidification and eutrophication. Depending on the requirements of the study, additional impact categories may be considered. Figure 2 presents the stages of the Life Cycle Impact Assessment (LCIA) according to the definitions of the impact categories.

According to Pizzol et al. [41], the great challenge of the LCIA methodology is to assess the potential impact using an applicable procedure, considering a common measurement unit and providing comparable data between impact categories. Another important point is the development of methods that consider global impacts and/or impacts relative to specific regions, such as specific countries: Canada, Europe, Japan and the United States. Thus, these methods do not necessarily reflect the situation of countries such as Brazil, which still does not have specific LCIA methods for the country’s environmental characteristics [27].

## 3. Life Cycle Assessment in Egg Production

In this review, the life cycle assessment tool addressed will be the environmental life cycle assessment. This is considered a widely used tool for assessing the intensity of resources used and product emissions from a supply chain perspective [42].

The ISO standardized framework [21,22] for LCA provides prescriptive guidance for characterizing inputs and emissions of materials and energy along product supply chains and for quantifying how these flows contribute to a variety of resources used, human health and potential environmental impacts.

On a global scale, researchers’ approach LCA as a constructive and quantitative technique, showing great interest in implementing LCA for the egg production process. Table 1 describes the current LCA studies, between the years 2018 and 2022, in the egg production process, its structure and considerations.

The studies presented in Table 1 predominantly highlight the relevance of the LCA and identify that the feeding of laying hens and the proper management of manure are the main contributors to the emission of greenhouse gases and impact the life cycle of the eggs. It was also possible to identify that these studies did not include the complete life cycle from the cradle to the retail of the eggs, only up to the pre-gate of production [20,43,44,45,46,47,48,49,50]. It should be noted that the most relevant impacts considered in the studies presented in Table 1 were: Acidification (kg SO_2_ eq.), Eutrophication (kg N eq.), Fossil Fuel Depletion (MJ surplus), Global Warming Potential (kg CO_2_ eq.), Ozone Depletion (kg CFC-11 eq.) and Ecotoxicity CTUe (comparative toxic unit equivalents).

The studies proposed by Cederberg et al. [62], Pelletier et al. [63] and Pelletier [42] analyzed the advances in egg production between the years 1990 and 2005 (Switzerland), 1960 and 2010 (USA) and 1962 and 2012 (Canada), respectively, through LCA. As a positive effect, it was found that, over the years, the footprint has been reduced, that is, the impacts on the environment in the production system are being rethought.

The Brazilian studies that are closest to the objectives proposed in this review are the references by Silva et al. (2014) [64] and Fernandes (2020) [65]. The authors [64] compared the environmental burden of two small-scale and large-scale broiler production systems in Brazil and two in France. The author [55] analyzed the environmental sustainability of different egg production environments. The research focused on analyzing the ambience of open, closed and alternative external warehouses.

In Brazil, there is also no specific norm for the intensive production of fresh eggs, but the intensive egg production system undergoes constant changes. The ABNT (Brazilian Association of Technical Standards) [52] considers only NBR 16437: 2016 Poultry—Production, classification and identification of free-range egg [53], paying attention only to the semi-extensive production of free-range eggs. Because it is an intensive production system with growing demand, studies are needed that clearly and accurately indicate the current impact of egg production on the environment, as every production process generates the depletion of natural resources, material flows, the cost of energy and increased emissions along the production chain [18].

### 3.1. Life Cycle Assessment Methods

According to Horne et al. [66], when defining the scope of the LCA, the limits of the system are determined, with the identification of the entire production process. Figure 3 illustrates the flow diagram of the egg production system.

The system boundaries for a study include all relevant material, energy and emission flows linked to all processes in the cradle-to-farm egg supply chain. This includes breeder, hatchery, pullet and layer facilities. As this is a start-to-finish assessment of the farm gate environmental life cycle (i.e., study of the environmental footprint) of conventional egg production, it is worth noting that the use/reuse/maintenance part will not be considered, since this is the “post-gate” analysis. This assessment of the life cycle of the egg production system does not consider the “post-gate” analysis [67]. Figure 4 represents the limits of the Life Cycle Assessment system for Egg Production, according to Turner et al. [44].

It is worth mentioning that transports between processes are also considered. The most used functional unit is the ton of eggs per unit of time. The functional unit and the limits of the system can be chosen as long as they are in agreement with the comparisons between the results [42]. Every unit involved in the production system must be included, as well as all component inputs and outputs, such as emissions and waste produced [40]. The analysis also does not include inputs and emissions associated with the production and maintenance of infrastructure, such as machinery and buildings. They typically make trivial contributions to the supply chain [42]. Figure 5 summarizes the Inputs/Outputs of the Egg Production Process, along with their respective (assigned) units. It is important to emphasize that this information is not immutable, showing only a suggestion of relevant inputs/outputs.

Midpoint impact categories and category indicators are employed in the impact assessment phase of the life cycle. Pelletier et al. [42] describe the impact categories in their study of LCA in the egg production system, considering the following as impact categories: Global warming (CO_2_ equivalence), acidification (SO_2_ equivalence) and eutrophication (PO_4_ equivalence), cumulative energy demand and water and land use.

It is important to highlight that the impact categories do not follow a specification and may vary according to the assessment carried out.

Allocation is a common strategy for solving multifunctionality problems in LCA, but the ISO 14040 and 14044:2006 [21,22] standard requires interpretations that are difficult to implement in practice. According to Pelletier et al. [58], three divisions may favor the allocation of an LCA, namely: the consequence and attributes of international data modeling approaches; adherence to a natural science-based approach; and, finally, a socioeconomic approach. The allocation of co-products is defined as products used in another economic activity. Unproductive chickens, for example, can be consumed (human food) or processed for animal feed, thus being destined. When they are not destined, they are sent for incineration and composting, being only discarded [57].

The choice of allocation methods is the target of criticism in LCA studies. Studies need to indicate how allocation systems are modeled, including which allocation procedure will be applied. The methods generally chosen by practitioners are economic allocations (based on economic value) and bulk allocations (based on reference weight or volume). The allocation factors are represented by the value of the product considered (according to the allocation unit chosen) and by the total value of the products considered in the system [68].

### 3.2. Life Cycle Assessment Analysis

An essential element in LCA practice is the distinction between foreground data and background data. Foreground data are considered the data of primary concern, and background data are delivered to the foreground system as aggregated datasets, where operations are not identified [27]. Foreground data for egg production are usually collected from the producer’s database. The corresponding values can also be considered as weighted averages of production (calculated values). Background system data will be required for integration with inventory data but modified where possible for Brazilian conditions.

As per ISO 14040 and 14044:2006 [21,22], data quality was assessed for foreground processes (i.e., egg production and layer manure management) as well as background processes. Bamber et al. [69] concluded that less than 20% of the LCA studies published between the years 2014 to 2018 reported any type of uncertainty analysis. Parameter uncertainty (i.e., uncertainty in inventory data) is most frequently reported, with 82% of studies, although other sources of uncertainty are considered equally important. Monte Carlo analysis was the most popular method, with 301 publications (61%) using it to propagate uncertainty results regardless of the LCA type.

### 3.3. Impact Assessment Methodologies

Currently, there are a considerable amount of software, developed by research centers, universities and companies around the world, that help in the development and execution of the LCA of different products and services. These computer programs facilitate the manipulation of the large and varied amount of data that an LCA requires. Table 2 lists some of the LCA software available on the market. It is important to emphasize that these are just a selection of software, and other software may be available.

OpenLCA [71] is an open-source software for Life Cycle Assessment (LCA) and Sustainability, developed in 2006 by GreenDelta [31]. The software is considered to be a data integrator that integrates the databases available from providers and networks. The Ecoinvent Life Cycle Impact Assessment (LCIA) set of methods is available through the openLCA Nexus [30]. Networks such as OpenLCA Nexus and the Ecoinvent database (versions 2.2 to 3.6) are identified as possible solutions to the impacts generated in production, allowing for the greater distribution and interoperability of data for life cycle assessments [74].

Developed by DSM animal nutrition and production experts [75], the Sustell Intelligence Platform [73] is a tool for the data entry, measurement and visualization of end-to-end results on environmental footprint. Based on a complete LCA, the software’s precise calculation allows for tangible and measurable improvements, from agricultural feed production to the final product such as broilers, dairy products, laying hens and pig fattening. The Sustell software provides data transparently to the producer, with the main objective of reducing the environmental footprint of a farm.

### 3.4. Life Cycle Interpretation

The life cycle assessment and interpretation analyses described by ISO 14040 and 14044:2006 [21,22] and recommended by UNEP/SETAC (United Nations Environment Program/Society for Environmental Toxicology and Chemistry) [19] result in criteria for three areas of protection: human health, ecosystem quality and natural resources. The definition of these areas aims to safeguard the values considered important for society. Based on the entire review presented here, it was possible to identify the main environmental impacts caused by egg production.

Along with calculations of the lifecycle environmental impacts of egg production, the authors [17,32,44,48,49] describe how a contribution analysis can determine impact hotspots across the entire egg production chain. It was also possible to define the expected critical points, with the feed and manure production processes being the most critical. The identification of hotspots can provide valuable information for industries and egg producers to provide targeted strategies for reducing the environmental impacts of egg production systems [58]. However, the LCA studies conclude their evaluation with precise, transparent and globally recognized results, resulting in the provision of the total environmental footprint (Kg CO_2_ eq) per product unit.

Recent studies (2022) on the assessment of the life cycle impacts of egg production in the European Union [74] have analyzed the four laying systems, namely: enriched cages, barns, free range and organic. The authors state that the composition of the feed and the handling of the manure are the factors that directly affect the total environmental impact of the eggs, regardless of the posture system adopted. They also conclude that organic eggs have more significant environmental impacts than conventionally produced eggs, due to the adaptation of laying hen diets. In the same year [44], contemporary Canadian egg production systems were evaluated during the transition from conventional cages to alternative housing systems. Feed formulations and different management systems between caged and cage-free pullet production systems were the main contributors to environmental impacts. The results indicated that conventional caged methods outperform other productions and that the transition from traditional cages can be negative for the environmental sustainability of Canadian egg production.

It is important to emphasize that all the studies presented in this review are not the only or exclusive ways to perform an LCA of a product or process. All studies generally provide an assessment guideline to better understand and address the impacts generated throughout its life cycle, along with the need to consider issues related to climate change and biodiversity, from a holistic perspective [84].

## 4. Conclusions

This review presents, in general terms, the Life Cycle Environmental Assessment (according to ISO 14040 and 14044:2006) [21,22] of an egg-producing farm, from cradle to gate, and the tools available for this purpose. The studies presented predominantly highlighted the relevance of LCA and identified that the feeding of laying hens and the proper management of management are the main contributors to the emission of greenhouse gases and other negative impacts on the environment in the life cycle of the eggs. These studies did not include the complete life cycle from the cradle to egg retail. Although chicken eggs are consumed worldwide as a valuable and inexpensive source of protein, there is an obvious lack of studies on the environmental performance of production. Brazil is considered a productive country but is in development. For this reason, most references addressed are international, which often does not match the Brazilian reality. A data gap was identified regarding accounting for inputs such as energy flows and natural resources, data on sustainability in different egg production systems (conventional and alternative) and the impact of each of these systems on the environment. One of the biggest challenges for researchers and producers is to obtain incentives for the use of flexible and transparent tools that can clearly and transparently translate their environmental footprint. In addition, it is expected that the references given as results will be useful in expressing future discussions about the impacts on each stage of the egg production chain.

## Figures and Tables

**Figure 1 animals-13-01479-f001:**
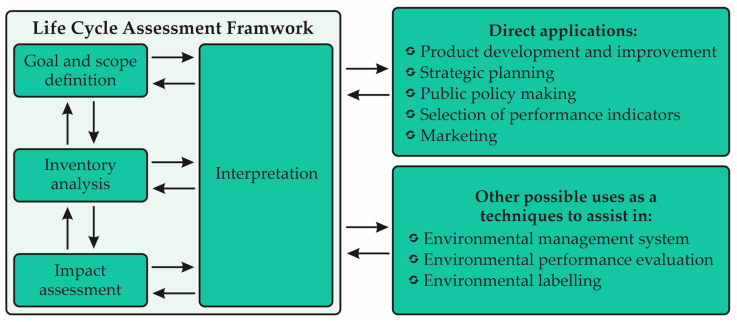
Life Cycle Assessment Framework, according to the ISO 14040 and 14044:2006 [21,22].

**Figure 2 animals-13-01479-f002:**
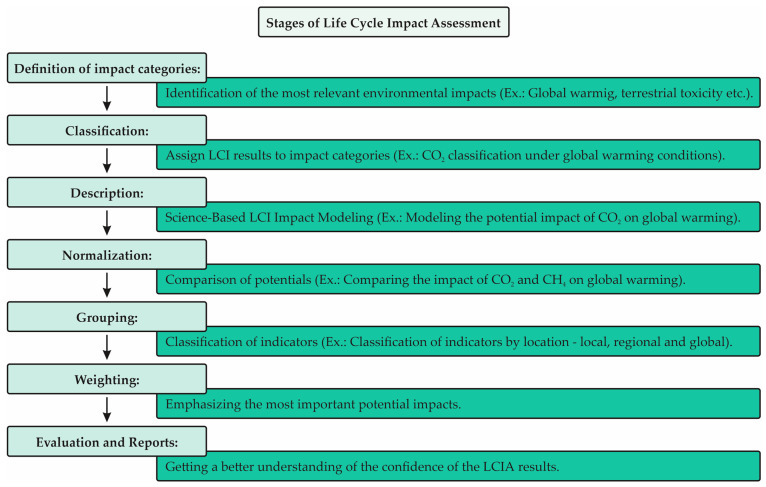
Stages of Life Cycle Impact Assessment (LCIA). Source: EPA [27]—Adapted by the author.

**Figure 3 animals-13-01479-f003:**
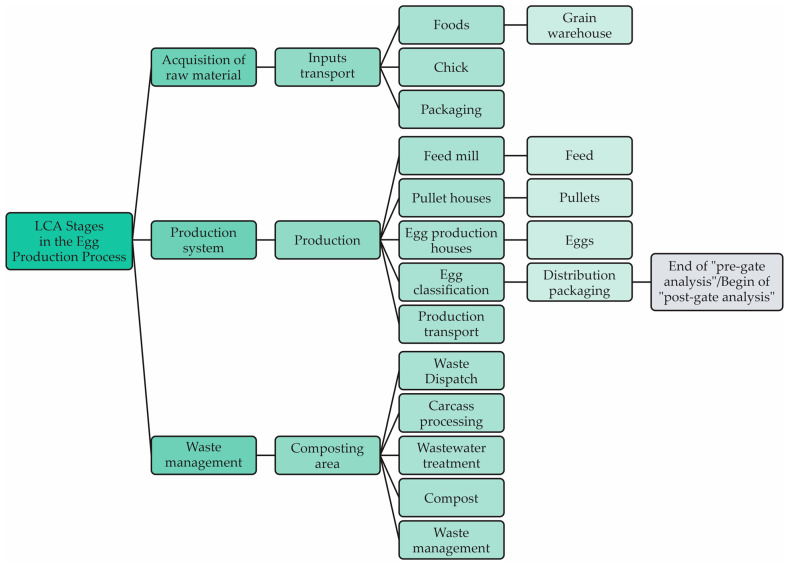
Flow diagram of the egg production system.

**Figure 4 animals-13-01479-f004:**
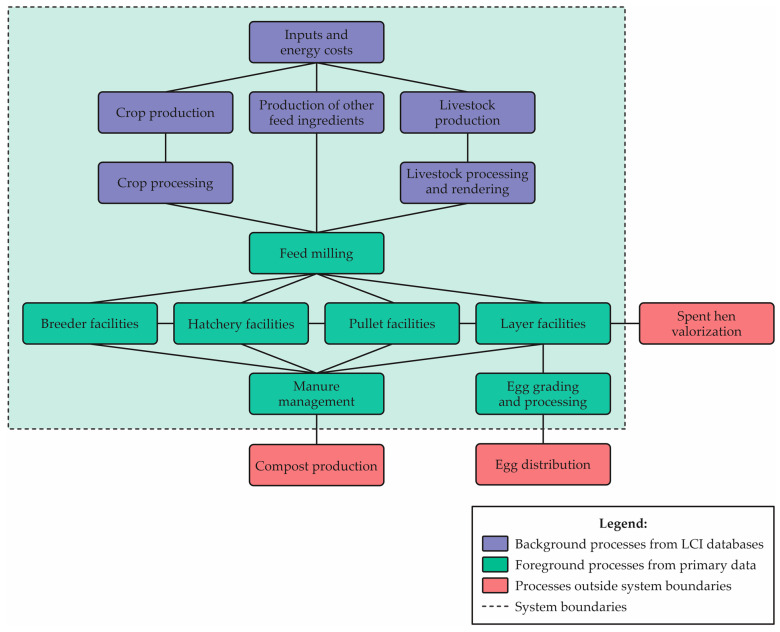
Limits of the egg production Life Cycle Assessment system. Source: Turner et al. [44]—Adapted by the author.

**Figure 5 animals-13-01479-f005:**
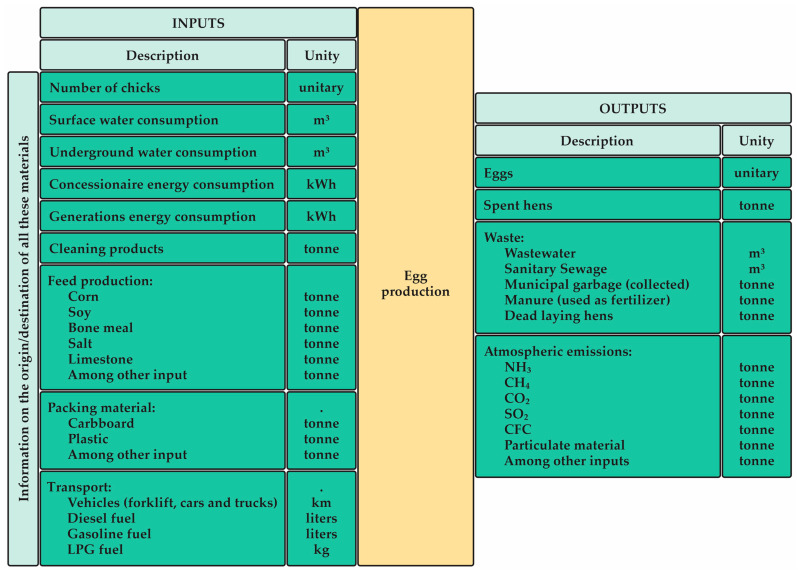
Input and Output analysis in the Egg Production Process (just a suggestion of relevant inputs/outputs).

**Table 1 animals-13-01479-t001:** Literature review on Live Cycle Assessment in the egg production process, published between 2018 and 2022.

Reference, Local	Article	Conclusions
Guillaume et al. (2022) [43], Czech Republic	Environmental Impacts of Egg Production from a Life Cycle Perspective	Feed composition and manure management are the factors with the greatest environmental impact, and the results suggest FCR ^1^.
Turner et al. (2022) [44], Canada	Life cycle assessment of contemporary Canadian egg production systems during the transition from conventional cage to alternative housing systems: Update and analysis of trends and conditions	Feed inputs are the largest contributors to the impact categories (18–84%), followed by pullet production and manure management (10–37% and 0.01–62%, respectively). Conventional cages had lower impacts than all non-organic systems.
Salami et al. (2022) [45], United Kingdom	Performance and environmental impact of egg production in response to dietary supplementation of mannan oligosaccharide in laying hens: A meta-analysis	^2^ MOS supplementation at 1.0 kg/ton improved the productive performance of laying hens and reduced the carbon footprint. Low and high ^3^ SBM diets reduced emissions by: dozen eggs (−0.02 and −0.03 kg CO_2_ eq); egg unit (−2.2 and −2.5 g CO_2_ eq); and kg of eggs (−0.04 and −0.04 kg CO_2_ eq).
Turner et al. (2022) [46], Canada	Environmental impact mitigation potential of increased resource use efficiency in industrial egg production systems	Potential reductions in pullet and feed consumption are up to 13.22%. The impacts are reduced by up to 17.27%.
Mitrovic et al. (2022) [47], Serbia	Assessment of Environmental Impacts from Different Perspectives—Case Study of Egg Value Chain System in Serbia	The productive chain of table eggs emitted 3.33 kg CO_2_ eq/kg egg, 29.01 MJ eq/kg, 17.76 g SO_2_ eq/kg and 27.79 g PO_4_ eq/kg. Eggs on farms had the greatest environmental impact due to the supply of feed for laying hens (74.94%) and the use of natural resources (24.42%).
Arulnathan et al. (2022) [48], Canada	Internal causality in agri-food Life Cycle Assessments: Solving allocation problems based on feed energy utilization in egg production	^4^ The ME model was used to quantify the allocation rates of eggs and chickens slaughtered in different systems. The egg allocation rate is between 82.6% and 97.5%. The co-product of spent chickens can be allocated up to 25% for net energy.
Ershadi et al. (2021) [49], Canada	Comparative life cycle assessment of technologies and strategies to improve nitrogen use efficiency in egg supply chains	Strategies and Acid Scrubber provide ^5^ NUE improvement options (15% and 13%, respectively). These strategies reduce acidification (35% and 21%) and eutrophication potential (26% and 16%), but they increase other impacts, such as energy consumption and the depletion of the ozone layer.
Tsai et al. (2021) [50], USA	Life cycle assessment of cleaning-in-place operations in egg yolk powder production	The ^6^ LCA was used to quantify the impacts of the different stages in the manufacture of powdered egg yolk. The total result obtained was 1.71 kg CO_2_ eq.
Li et al. (2021) [51], Canada	Net zero energy barns for industrial egg production: An effective sustainable intensification strategy?	A facility with ^7^ NZE poultry housing infrastructure will provide environmental benefits over time. Lifecycle environmental impacts of eggs are 0.89–64.82% lower in the NZE shed.
Costantini et al. (2021) [52], Italy	Environmental sustainability assessment of poultry productions through life cycle approaches: A critical review	One hundred and fifty-five studies were imposed, of which forty-seven were reviewed. The agricultural phase weighs heavily on the impact of the finished food product. However, feed consumption and waste management are primarily responsible for the impacts generated.
Kanani et al. (2020) [53], Canada	Waste valorization technology options for the egg and broiler industries: A review and recommendations	^6^ LCA studies represent only 4% of the literature in this review. Currently, there is no link between the academic literature and the adoption of technologies for the valorization of poultry waste. Therefore, it is essential to carry out detailed studies (regionalized) to determine and understand the environmental resources and the waste generated.
Costantini et al. (2020) [54], Italy	Investigating on the environmental sustainability of animal products: The case of organic eggs	Feed supply is the main access point (49% to 87%) for all impact categories (1.56 kg CO_2_ eq/kg). The impact is less than that for conventional eggs.
Gunnarsson et al. (2020) [55], Sweden	Systematic Mapping of Research on Farm-Level Sustainability in Egg and Chicken Meat Production	The literature between the years 2000 and 2020 resulted in a mapping: only three articles covered the three dimensions of sustainability; ten addressed aspects of economic sustainability; eighteen addressed aspects of environmental sustainability; and twenty-three addressed aspects of social sustainability.
Oryschak et al. (2020) [56], Canada	Reconsidering the contribution of Canadian poultry production to anthropogenic greenhouse gas emissions: returning to an integrated crop–poultry production system paradigm	The carbon footprint is considered a discourse around climate change policy, but the exclusion of carbon fixation perpetuates a mistaken assumption that livestock is a net contributor to the ^8^ GHG emissions problem by replacing part of a solution.
Fritter et al. (2020) [57], Canada	A survey of Life Cycle Inventory database implementations and architectures and recommendations for new database initiatives	For the development of new ^9^ LCI database features, the format, nomenclatures, third-party providers, third-party initiatives and technical implementation are recommended.
Estrada-González et al. (2020) [19], Mexico	Decreasing the Environmental Impact in an Egg-Producing Farm through the Application of LCA and Lean Tools	The climate change category is a hotspot in egg production, with emissions of 5.58 kg CO_2_ eq/kg per egg produced.
van Hal et al. (2019) [58], Netherlands	Accounting for feed-food competition in environmental impact assessment: Towards a resource efficient food-system	Using ^10^ LCF economic allocation reduced ^11^ GWP by 48–58%, ^12^ EU by 21–37%, ^13^ LU by 34–47% and ^14^ LUR by 32%. Using ration-based allocation, the impact per kg of egg was further reduced by 54% for GWP, 38% for EU, 94% for LU and 88% for LUR.
Vetter et al. (2018) [59], United Kingdom	The potential to reduce GHG emissions in egg production using a GHG calculator—A Cool Farm Tool case study	The highest GHG ^8^ emissions come from feeding, followed by transport and manure management. The results show that the average GHG emissions decreased over the three years of the study by almost 25%.
Abín et al. (2018) [60], Spain	Environmental assessment of intensive egg production: A Spanish case study	Land use was the most prominent category, followed by terrestrial ecotoxicity and freshwater ecotoxicity. The carbon footprint of egg production was calculated to be 2.66 kgCO_2_ eq per dozen eggs.
Pelletier et al. (2018) [61], Canada	Sustainability in the Canadian Egg Industry—Learning from the Past, Navigating the Present, Planning for the Future Nathan	The challenges presented are aimed at egg producers. Acquiring practical knowledge, transitioning management and housing systems or not and analyzing the economic values of new systems, among other points, must be evaluated. This analysis can identify preferred paths, potential pitfalls and outstanding interdisciplinary research questions.

^1^ FCR (Feed Conversion Ratio); ^2^ MOS (Mannan Oligosaccharides); ^3^ SBM (Soybean Meal); ^4^ ME (Metabolizable Energy); ^5^ NUE (Nitrogen Use Efficiency); ^6^ LCA (Life Cycle Assessment); ^7^ NZE (Net Zero Energy); ^8^ GHG (Greenhouse Gas); ^9^ LCI (Life Cycle Inventory); ^10^ LCF (Low-Cost Feed); ^11^ GWP (Global Warming Potential); ^12^ EU (Energy Use); ^13^ LU (Land Use); ^14^ LUR (Land Use Ratio).

**Table 2 animals-13-01479-t002:** Software available for life cycle assessment, indicating providers and countries where they were developed (only software suggestions).

Tools	Provider	Country
SimaPro [70]	Pré-Sustainability	Netherlands
OpenLCA [71]	GreenDelta	Germany
GaBi [72]	Sphera	Germany
Umberto [73]	IPoint	Germany
Ecodesign Studio [74]	Altermaker	France
Air.e LCA [75]	Solidforest	Spain
ECOSPEED Scout [76]	ECOSPEED Climate Software Solution	Switzerland
EarthSmart [77]	EarthShift Global	USA
Ecodex [78]	Selerant	USA
PLACES [79]	The Circulate Initiative	USA
LCA4Waste [80]	ETH Zurich	Switzerland
GREET [81]	Argonne National Laboratory	USA
KCL-ECO [82]	KCL Piloting Knowledge	Finland
Sustell [83]	DSM	Netherlands

## Data Availability

The data presented in this study are available upon request from the corresponding author.

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
