# Peer review of "Life Cycle Assessment Project for the Brazilian Egg Industry"

_animals, 2023, doi:10.3390/ani13091479_

Round 1

Reviewer 1 Report

Dear Authors,

I am grateful for your manuscript. The subject is important - especially nowadays - when changes in livestock production are inevitable and the emphasis is on environmental protection.

Your manuscript is prepared carefully and precisely.

However, it is the reviewer's role to analyse the manuscript in detail and add some comments that are likely to increase the value of the manuscript.

Line 49 – Please explain why world wide egg production in Brazil was lower in 2021

Line 54 - Which countries? EU countries? Please give some examples.

Line 55 - IBGE - expand abbreviation. Please, check carefully in entire manuscript if every abbreviation has added full name.

Line 121 – Please use full name: ISO 14044:2006 – correct the entire manuscript. I believe it will be more readable.

Line 122 - Inverted commas not needed. Please explain what kind of footprint you mean. Carbon footprint? It is important to understand the aim of your manuscript.

Line 279 - Why have you made literature review only between the years 2020 and 2021? I recommend add information from last 5 years. In my opinion, showing information from 2020 to 2021 is too short period. I also recommend add information in the text how it looks in USA and European Union and compare information with Brazil.

3.4. Life Cycle Interpretation - And what does it look like in the European Union? I am missing information about this.

Figures comment: I recommend changing the colours of the figures to lighter colours. In my opinion, a lighter background will be more readable.

I recommend accept the manuscript after minor revision.

Author Response

Dear Reviewer,

We, the authors, are immensely grateful for all your consideration and your help in improving our manuscript. We will respond to all your comments and suggestions. Our responses are indicated in italics below and all changes are marked in gray in the manuscript.

Best regards,

The authors.

Reviewer 2 Report

Dear authors,

the manuscript provides a review of life cycle assessment for Brazilian Egg Industry. It is interesting because it addresses a gap of knowledge and will probably help to encourage stakeholders in carrying out such analyses. For me this review may function as a guideline and a groundwork for further discussions and developments and for promotion environmental sustainability in the Brazilian egg industry. Please see below some suggestions to improve the review:

Line 17+18: “on potential environmental impacts and their impact categories”

For me a LCA does not provide information about impacts AND impact categories but about the impacts separated into different categories. Therefore I would clarify this relation in this sentence.

Line 22-24: “covering all impact categories, being …”

As you are writing “all” for me the reader gets the idea that these impact categories are like the only right solution. I would highlight that these categories are one possible classification and that the ISO 14044 allows modifications.

Line 25+25: “socio-environmental”

The results of a LCA only provide data about environmental impacts. The mention of “socio” might be misleading here.

Line 39: “based on ISO standards. 14040 and …”

For me the dot does not belong here

Line 42:

I would add “LCA” to the keywords

Line 45/Reference 1:

Please check the link again. For me with “QI” at the end I am reaching information about the gross production, information about the world population are available with “OA” at the end

Line 48+49:

I would write both numbers with the equal amount of digits

Line 50/Reference 2:

Please check the writing of the reference. The surname is ‘Van Horne’, the year of the publication should be written in bold and the article is written on page 12 only, not page 1-393. And as I check for this details I wondered what information you took from this reference?

Line 51+52: “occupying the sixth position in the world ranking”

To clarify this statement I would add a year

Line 52-54/Reference 5+6:

Reference 5+6 are not mentioned in the text

Line 55: “IBGE”

The abbreviation should be defined first

Line 57: “… quarter of 2021 Despite the drop …”

Missing dot

Line 83: “National Health Surveillance Agency 82 (ANVISA) [13], National Institute of Metrology, Quality and Technology (INMETRO)”

I would end this enumeration with an “and” in-between

Line 121:

Here and below you mention the ISO 14044 only although LCA studies are bases on the ISO 14040 and the ISO 14044 (as you mentioned in the simple summary already). I would clarify at this point that both ISO are relevant and that in further sentecnces only the more accurate ISO 14044 (maybe also specify it as ISO 14044:2006 like in the simple summary here) is named

Line 122: “and analyze …”

For me the ‘environmental impact’ would be something quantifiable which was not addressed in this review. Maybe you can clarify this last statement writing that you analyze studies dealing with the environmental impact or the system of the environmental impact in general

Line 130: “environmental and potential impacts”

could be “potential environmental impacts”

Line 141+142: “The results ..”

Shouldn’t it be “The results of a product or service LCA promote …”?

Line 145-147: “These results will allow …”

This sentence implies that a higher efficiency is the only way to reduce environmental impacts which is not the case. Of course it is one way but a low efficiency does not always correlate with high environmental impacts

Line 173: “life cycle event (LCI)”

LCI is the abbreviation for life cycle inventory (see line 160)

Line 209+210: “six hundred and thirty-two”

Should be 632

Line 268-274: “e-LCA”

Isn’t e-LCA equal to LCA? If not, please highlight the difference

Line 279/Table 1:

VSM and NUE not explained;

“The production of fresh eggs was evaluated LCA.” is an incomplete sentence

“1.56 kg CO eq2•kg-1”, please check the unit again, I think I should be CO2 eq•kg-1

Line 288: “[20,43,44,45,46,47,48,49–50]”

Should be [20,43–50]

Line 305/Figure 3:

ETE not defined

Line 313: “the Limits of …”

Should be “the limits of …”

Line 315/Figure 4:

Please review the lines in this diagram again carefully.

For me this line should be added: 1) from “crop production” to “crop procession”, 2) from “livestock production” to “livestock processing and rendering”, 3) from “crop processing” to “feed milling”

And this lines are questionable: 1) from “production of other feed ingredients” to “crop processing”, 2) from “production of other feed ingredients” to “livestock processing and rendering”, 3) from “feed milling” to “hatchery facilities”, 4) from “hatchery facilities” to “manure management”

Line 325+326:

I would mentioned again here (and/or in the description of figure 5) that this is not set in stone and shows only a suggestion of relevant inputs/outputs

Line 327/Figure 5: “EGGs production”

Should be “egg production” or “production of eggs”

Line 331-334:

I would highlight here that these impact categories are only an example a not a specification

Line 336-343:

Maybe you can mention recognized allocation systems here like the economic allocation or the allocation by mass

Line 366/Table 2:

I would add the information to the description of the table that this is only a selection (as there is also other software available)

Line 369-383:

Why do you describe these tools in detail? How do you selected them from the others?

Line 385:

UNEP and SETAC not defined

Line 396-398:

The last sentence of this paragraph suggest that the impact of all categories is expressed in CO2 eq which is not the case. I think this should be clarified here.

Line 400+406: “by the authors”

It might be difficult for the reader to identify which authors are meant. Those of the reference or those from the review itself? It might be helpful to clarify that

Line 414:

I would add a note that this is not like “the only right way” to perform a LCA. Maybe it would be helpful to declare this review as a “guideline” (open for adjustments)?

Line 418: “greenhouse gases and impact”

Should be “greenhouse gases and other negative impacts on the environment”

References:

Please review the writing of the references and its format. The name of the authors must not be written in capital letters, the year need to be written in bold, …

Kind regards

Author Response

Dear Reviewer,

We, the authors, are immensely grateful for all your consideration and your help in improving our manuscript. We will respond to all your comments and suggestions. Our responses are indicated in italics below and all changes are marked in yellow in the manuscript.

Best regards,

The authors.

Round 2

Reviewer 2 Report

Dear authors,

well done. I think this manuscript is now ready to publish.

Kind regards